# Consumption of Ultra-Processed Foods Is Associated with Free Sugars Intake in the Canadian Population

**DOI:** 10.3390/nu14030708

**Published:** 2022-02-08

**Authors:** Virginie Hamel, Milena Nardocci, Nadia Flexner, Jodi Bernstein, Marie R. L’Abbé, Jean-Claude Moubarac

**Affiliations:** 1Department of Nutrition, Faculty of Medicine, University of Montreal, Montreal, QC H3T 1A8, Canada; milena.nardocci.fusco@umontreal.ca (M.N.); jc.moubarac@umontreal.ca (J.-C.M.); 2Centre de Recherche en Santé Publique, University of Montreal, Montreal, QC H3N 1X9, Canada; 3Department of Nutritional Sciences, Temerty Faculty of Medicine, University of Toronto, Medical Sciences Building, Room 5368, 1 King’s College Circle, Toronto, ON M5S 1A8, Canada; nadia.flexner@mail.utoronto.ca (N.F.); jodi.bernstein@mail.utoronto.ca (J.B.); mary.labbe@utoronto.ca (M.R.L.)

**Keywords:** ultra-processed food, free sugars, public health, nutrition, population diet

## Abstract

Excess sugar consumption can lead to noncommunicable diseases (NCDs) such as type 2 diabetes. Increasingly, ultra-processed foods (UPF) are suspected to be great contributors to free sugars intake in the population’s diet. Thus, the aim of this study was to investigate the association between UPF consumption and free sugars intake in the Canadian population. We used data from one 24 h-recall of the nationally representative 2015 Canadian Community Health Survey–Nutrition (CCHS). Food items were classified according to the NOVA system, and to estimate free sugars intake, we used the University of Toronto’s Food Label Information Program (FLIP) 2017 database. Results: Almost half of the population’s energy intake (45.7%) came from UPF. On average, 221.5 kcal/day came from free sugars, and most of these calories (71.5%) came from UPF. Public health policies aiming to decrease consumption of UPF should be a priority considering their important contribution to sugar intake in the population.

## 1. Introduction

High intake of free sugars is associated to excess body weight and obesity that, in turn, increases the risk of obesity-related diseases including type 2 diabetes [1,2,3]. The Canadian Heart and Stroke Foundation [4] and the World Health Organization [3] (WHO) recommend limiting free sugars intake to less than 10% of total energy intake, and ideally less than 5% to minimize health risks. While the 2019 Canadian Food Guide recommends limiting food high in sugars, such as candies, cookies and cakes, sweetened juices and drinks, sweetened milk products, amongst others [1,3,4,5], the free sugars levels of these products are not easy to identify, given that it is not declared on the Canadian Nutrition Facts table. 

Free sugars are defined as monosaccharides and disaccharides added to foods by the manufacturer, cook, or consumer, plus the sugars that are naturally present in honey, syrups and fruit juices [3]. Based on the 2015 Canadian Community Health Survey data, free sugars account for 13.3% of the daily calories consumed by Canadian adults [6], which exceed the WHO recommendation of less than 10%. Liu and al. [6] found that the main source of free sugars in Canada are desserts and sweets followed by beverages. Considering this high free sugars consumption, it is imperative to find intervention strategies to reduce free sugars intake in the Canadian population.

To do so, two general strategies exist. One strategy is to reformulate food products to reduce problematic nutrients such as free sugars [7]. However, reformulation of food products aiming at reducing free sugars might lead to unintended consequences, such as replacing sugars with refined starches or fats, which could increase caloric content of the product [8]. Furthermore, by focusing on the nutrients-to-limit (sugar, fat, and salt), reformulation may not result in more nutritious products. For instance, the use of artificial sweeteners to replace sugar is common, but is also suspected to be linked to adverse health outcomes [9]. Reformulation can also lead to the legitimation and promotion of products that are posing a lesser risk for health, instead of limiting their consumption [10]. 

The second strategy to reduce free sugar intake is to reduce consumption of ultra-processed foods (UPF), as those products have been identified as important sources of free sugars [11]. UPF are industrial formulations made of refined substances (including free sugars, sodium, and fats) and additives with little or no whole foods. They include carbonated drinks, fruit juices and drinks, candies, snacks, reconstituted meats, sauces and dressing, and many other products [12]. Several countries have conducted population-based diet studies and have shown that UPF products are the main source of free sugars and among these are Australia, United Stated, Canada, and United Kingdom [11,13,14,15]. For instance, in the US, UPFs accounted for 57.9% of daily energy intake in 2009–2010 and was the greater contributor to sugar intake representing 89.7% of the energy intake from added sugars (added sugars do not include natural occurring sugar present in fruit juices and syrups like honey) (Martinez Steele et al., 2016). Furthermore, each increase of five percentage points in proportional energy intake from UPF increased the proportional energy intake from added sugars by one percentage point in both unadjusted and adjusted models for co-variates [13]. Adjusted quintiles analyses of the 2004 Canadian Community Health Survey (CCHS) suggest that free sugars intake is directly related to consumption of UPF: the contribution of free sugars to total calories rises from 7.7% to 19.4% when comparing Canadians consuming the lowest quintile of UPF (23.5% of total daily calories) to those consuming the most (76.2% of total daily calories) [11].

Using the 2015 CCHS-Nutrition survey data, this study investigates the association between UPF consumption and free sugar intake in the Canadian population. Based on prior studies, our hypothesis is that UPF intake is directly and strongly associated to free sugars intake in Canada. It is important to verify this assumption to develop the most effective interventions aimed at reducing free sugars intake and prevent health issues associated with it. 

## 2. Materials and Methods

### 2.1. Data Source and Study Sample

We used data from the nationally representative 2015 Canadian Community Health Survey–Nutrition (CCHS) Public Use Microdata File (PUMF) [16]. The general CCHS program consists of series of cross-sectional surveys aiming at providing health information for the Canadian population. The 2015 CCHS-Nutrition survey’s main goal is to provide general information about the dietary intake of Canadians [16]. The target population included all individuals aged 1 year and above living in private dwellings in the 10 Canadian provinces. The survey excluded full-time members of the Canadian Forces, individuals who lived in the Territories, on reserves and other Indigenous settlements, in some remote areas, and in institutions (e.g., prisons or care facilities) [16].

The survey included interviews conducted in person at the participants’ homes. For children aged 1 to 5 years, interviews were conducted with a proxy only (parent or guardian); for children aged 6 to 11 years, interviews were conducted with the child’s parent or guardian in the presence of the child; and children aged 12 years and above were interviewed on their own. The interview included a 24 h-diet recall and questions aiming to support the interpretation of the 24 h-recall. The United States Department of Agriculture Automated Multiple-Pass Method (AMPM) adapted for the Canadian context was employed to maximize the participants’ recall of foods and drinks consumed in the previous 24 h of the day of the interview [16]. The recommended five steps of the AMPM were followed for this survey, namely the (1) quick list, (2) forgotten foods, (3) time and occasion, (4) detail cycle, and (5) final review. A random subset of individuals (about 35%) was invited to complete a second 24 h-recall, 3 to 10 days after the first interview, on a different day of the week, by telephone [16]. In total, 20,487 individuals took part in the survey and the response rate was 61.6% [16]. 

For this study, only the first 24 h-recall was used to determine the average intake at the population level [17]. All study participants were included in this study except children aged less than 2 years (*n* = 372), and individuals who did not consume any calories in the previous day of the survey (*n* = 12). The final analytic sample was 20,103 participants and socio-demographic characteristics of the sample are presented in Table 1. Data analyses were performed using the Public Use Microdata File (PUMF) which excludes some data (e.g., urban/rural zone) in order to protect the confidentiality of the respondents [16]. 

### 2.2. Food Classification According to Type of Processing

All foods and drinks reported in the first 24 h-recall were classified according to the NOVA classification, which groups food according to the nature, extent, and purpose of the industrial processing [18]. Foods were classified into four distinct groups: (1) unprocessed and minimally processed foods, such as frozen fruits and vegetables, plain milk, pasta, and flour; (2) processed culinary ingredients, which includes oils, butter, sugar, and salt; (3) processed foods, such as canned vegetables, canned fish, fruits in syrup, cheese, and freshly made artisanal breads; and (4) ultra-processed foods (UPF) which are formulations made mostly or entirely from substances derived from foods and additives, with little if any whole foods from NOVA group 1. Examples of UPFs are mass-produced industrial breads and buns, reconstituted meat products, commercial fruit juices and fruit drinks, and confectionary (e.g., chocolate, candies, desserts). More details on how each food and drink were classified according to the NOVA classification were described in detail elsewhere [2]. 

### 2.3. Free Sugars Content of Foods

To estimate free sugars content, we used the University of Toronto’s Food Label Information Program (FLIP) 2017 database. FLIP is a database that provides nutritional information for 17,671 unique food and beverage products found in the Canadian market [19]. For this study we used data that provided estimates on free sugars content per 1 g of food and beverage. More details about the FLIP 2017 database procedures are described elsewhere [19].

For all NOVA group 3 (processed) and group 4 (UPF products), each food and drink reported in the 2015 CCHS was matched with corresponding FLIP products, to obtain an estimate of free sugars per gram consumed. Data access to the free sugars database was granted by the University of Toronto to University of Montreal under the contract (503154-subgrant1). Average values were calculated for CCHS foods that could be matched with several FLIP products.

### 2.4. Data Analysis

First, we calculated total energy intake and energy intake from free sugars for each NOVA food groups, both absolute (kcal/day) and relative to energy intake (% of total energy intake). We then assessed the mean relative energy intake from UPF and from free sugars across quintiles of UPF consumption (% energy intake). Relative energy intake was calculated using the population ratio approach (total intake of the category of UPF for the entire population over total energy intake for the entire population). This approach better reflects usual intake at the population level as compared to calculating the proportion for each individual and averaging it to obtain a mean for the sample [20]. The first quintile represents the lowest UPF relative energy intake and the fifth the highest. 

Next, the WHO recommendation limits of daily intake of free sugars [3] were used to verify the association between average UPF intake and the prevalence of excessive free sugars intake. For this, the prevalence of participants consuming more than 5% and more than 10% of total energy intake from free sugars across quintiles of UPF consumption (% of energy intake) was assessed. Based on prior studies, prevalence ratios were adjusted for sex (men; women), age (years, continuous), household income adequacy (grouped into quintiles), household educational attainment (grouped as less than high school; high school; trade, college, or CEGEP; or university diploma or above), and immigrant status (Canadian-born or immigrant) [21].

All analyses were conducted in SAS 9.4 and applied survey sampling weights provided by Statistics Canada to account for the complex sampling design and unequal probability of selection. Bootstrap weights were used to calculate robust standard errors using the Balanced Repeated Replication (BRR) method, as recommended by Statistics Canada [16]. Alpha level was set at 0.05. In order to account for the complex survey design, adjusted prevalence ratio was calculated using Cox proportional hazards models for complex surveys (PROC SURVEYPHREG in SAS 9.4), assigning an equal time of follow-up to all individuals [22].

## 3. Results

### 3.1. Distribution of Total Energy Intake by NOVA Food Groups

In 2015, the average daily energy intake in Canada was 1795.9 kcal, and almost half (45.7%) came from UPF (Table 2). Unprocessed and minimally processed foods contributed to 39.7% of total energy intake, processed culinary ingredients 6.9%, and processed foods 7.7%. 

### 3.2. Distribution of Energy Intake from Free Sugars by NOVA Food Groups

On average, 221.5 kcal/day came from free sugars, and most of these calories (71.5%) came from UPF. Processed culinary ingredients contributed to another 20.6%, while the contributions of unprocessed and minimally processed foods (6.0%), and processed foods (1.9%) were considerably lower (Table 2). The average relative content of free sugars in UPF (19.3% of kcal) was six-fold higher than in processed foods (3.1%), and almost three-fold higher than in unprocessed and minimally processed foods and processed culinary ingredients grouped together (7.0%) (Table 2).

### 3.3. Association between Consumption of Ultra-Processed Foods and Free Sugars Intake

The relative daily energy intake from UPF (% total energy intake) ranged from 19.4% (quintile 1) to 76.4% (quintile 5). Across quintiles of UPF consumption, the intake of free sugars was 8.1% in the lowest quintile and 16.8% in the highest quintile (Table 3). The prevalence of participants consuming more than 10% of total energy from free sugars was 53.5% in the total population, and across the quintiles of UPF consumption, this prevalence was 28.4% in quintile 1 and 73.1% in quintile 5. Similarly, the prevalence of participants consuming more than 5% of total energy from free sugars was 82.1% in the total population, and across the quintiles of UPF consumption, this prevalence was 65.3% in the first quintile (quintile 1) and 91.7% in the last quintile (quintile 5).

## 4. Discussion

This study supports the evidence that excessive consumption of free sugars by Canadians is quite important [6,11]. Our study found that overall, 53.5% and 82.1% of Canadians are not meeting WHO’s recommendation to eat less than 10% and 5% of total energy intake from free sugar, respectively. Moreover, it clearly demonstrates that the main source of free sugar is UPF products, and that high consumption of UPF products contributes to a higher consumption of free sugar in the Canadian population. Indeed, almost three quarters of the daily calories consumed as free sugar comes from UPF. The consumption of free sugars is 8% in the lowest quintile and almost 17% in the highest quintile of UPF consumption. Only Canadians that consume the least amount of UPF are consuming less than 10% of their daily calories from free sugars, which is recommended by the WHO [3]. As a secondary target, processed culinary ingredients contribute to about 20% of the daily calories from free sugars and therefore their use in preparing and cooking food should be examined carefully. Thus, eating UPF products and using a lot of sugar in homemade dishes seem to lead to the overconsumption of free sugars in the Canadian population. Our results confirm prior evidences from studies based on food expenditure data [23] and the 2004 CCHS surveys [11]. Using data from the 2004 CCHS–Nutrition, Moubarac et al. (2017) [11] compared the nutrient profile of the fraction of the diet made up of UPF with the same profile of the fraction made up of non-ultra-processed foods and showed that the fraction of UPF contained 250% more free sugars. Moreover, the study showed that in the highest quintile of the dietary share of UPF, sugar intake contributed with 19.4% of the total energy intake, which is slightly higher than the results from the present study (16.8%). 

Our results are also highly consistent with studies performed in other high-income countries based on national dietary intake surveys and using comparable methods. In the United Kingdom, Rauber et al. [15] showed that UPF accounted for 56.8% of total energy intake and 64.7% of total free sugar intake in the UK diet. In the United States, in a study about added sugar only, Martinez et al. [13] showed a strong positive linear association between the dietary contribution (% of energy intake) of UPF and the dietary content (% of energy intake) in added sugars: authors found that the contribution of added sugar to total energy intake was 7.5% in the first quintile and 19.5% in the fifth quintile. Similar to our study, one calorie out of five coming from UPF are added sugars, which would probably be higher if natural sugar present in fruit juices and syrups would be included in the analysis. Similarly, a positive linear association was found between quintiles of UPF consumption and both the average intake of free sugars and the prevalence of excessive free sugar intake in a nationally representative sample in Australia [14]. 

Our study has some strengths. We used a large nationally representative sample of the Canadian population. We also used the NOVA classification which is a validated tool for classifying foods according to the type of processing and has been used in several studies across the world. Our study also uses recent market data from the FLIP database (2017) to estimate free sugar content of foods and drinks Canadians consume.

Some limitations should be considered when interpretating our results. First, data from the 2015 CCHS had a prevalence of under-reporters of 7.5% higher in 2015 compared with the CCHS from 2004, while the prevalence of over-reporters was 7.4% lower [24]. This indicates that the relative free sugar energy intake from UPF could have been higher and even more similar to Moubarac’s [11] results. Moreover, 24 h recalls are subjected to potential bias, such as recall and social desirability that can underestimate real consumption of UPF. Also, we did not perform subgroup analysis by age, sex/gender, or ethnicity, which would have been relevant to identify public health intervention specific for different subgroup populations. However, based on prior studies, we believe the association between UPF and free sugar intake to be consistent across all segments of the population given that UPF intake is relatively high for all socioeconomic groups. However, as younger individuals consume more sugar drinks and foods than older individuals, we believe that the association between free sugar and UPF intake would be stronger for younger individuals [2]. Also, while First Nations people were not included in our analysis, a previous study conducted with First Nations on reserves reported that free sugars intake raised as the proportion of UPF increased in the diet [25].

This study suggests that encouraging the decrease of UPF consumption would be an efficient way to decrease free sugar intake and to comply with current dietary recommendations (i.e., eating less than 10% of total energy intake from free sugars). Limiting UPF consumption could also be a way to improve the overall quality of diets, by increasing consumption of foods that are fresh or minimally processed which have an overall healthier nutritional profile [11]. Given the growing body of evidence suggesting adverse effects of high consumption of UPF [26,27,28], some governments have taken the lead by recommending limiting UPF in their food guide [29,30]. Health Canada also refers to highly processed food in its food guide and suggests limiting their consumption in a way to reduce salt, sugar, and fat consumption [5]. However, there are currently no tools or ways to identify those products high in sugar for the consumer. Furthermore, reformulation of foods high in sugar may lead to more use of artificial sweeteners and/or an overconsumption of reformulated, less sugary products that may in the end lead to overconsumption of free sugars [10]. As mentioned previously, there are also concerns that the reduction of sugar in food products is offset by an increase in starch, which has the same energy density as sugar, and it can result in an increase in fat which is used as a substitute. So far, reformulation of sugary products in Canada lead to mitigate results [8]. Finally, industry-led reformulation data could be a way to delay implementation of effective nutrition public health policies aiming at improving the population’s diet [31], thus, it is important to count on independent monitoring of the food supply. 

While soda taxes have been implemented in some countries like Mexico [32] and United Kingdom [33] to decrease sugar intake, some argue that these kinds of taxes should be extended to sugary foods as well [34]. The taxation of both beverage and UPF could help the Canadian population to reduce free sugar intake. Public health policies aiming to decrease consumption of UPF should be a priority given that these products are abundant, often cheap, and highly advertised. 

## Figures and Tables

**Table 1 nutrients-14-00708-t001:** Sample characteristics (*n* = 20,103).

Variable	*N* (Weighted)	Mean or %	Standard Error (SE)
Sex, %			
Male	9554	49.3	0.17
Female	10,549	50.7	0.17
Age (years), mean	20,103	40.6	0.20
Household income adequacy, %			
Quintile 1	4125	19.9	0.65
Quintile 2	4120	20.3	1.16
Quintile 3	4362	20.4	1.64
Quintile 4	3729	19.6	0.61
Quintile 5	3745	19.8	2.54
Missing	22	-	-
Household education, %			
Less than high school	1769	6.2	0.44
High school	3712	16.4	0.67
Trade, college, CEGEP	7530	37.4	0.74
University diploma	7050	40.1	1.22
Missing	42	-	-
Immigrant status, %			
Non-immigrant	16,706	76.1	3.94
Long-term immigrant	2204	16.8	3.57
Recent immigrant (<10 years)	1109	7.1	0.53
Missing	84	-	-

**Table 2 nutrients-14-00708-t002:** Total energy intake and energy intake from free sugars by NOVA food groups among Canadians 2 years or older (*n* = 20,103).

Absolute Energy Intake (kcal/day)				Relative Energy Intake (% Total Energy Intake)			Absolute Energy Intake from Free Sugars (kcal/day)			Relative Energy Intake from Free Sugars (% Total Energy Intake from Free Sugars)			Relative Content of Free Sugars (Energy Intake from Free Sugars by Total Energy Intake)		
		95% CI			95% CI			95% CI			95% CI			95% CI	
Groups ^1^	Mean	Lower Limit (LL)	Upper Limit (UL)	Mean	LL	UL	Mean	LL	UL	Mean	LL	UL	Mean	LL	UL
NOVA 1	712.3	692.0	732.6	39.7	38.6	40.8	13.3	12.5	14.1	6.0	5.6	6.4	1.9	1.7	2.0
NOVA 2	124.0	120.0	128.0	6.9	6.7	7.1	45.5	43.4	47.6	20.6	19.7	21.4	36.7	35.5	38.0
NOVA 3	139.0	129.9	148.1	7.7	7.2	8.2	4.2	3.7	4.8	1.9	1.7	2.1	3.1	2.7	3.4
NOVA 4	820.5	791.9	849.1	45.7	44.3	47.1	158.4	153.8	163.1	71.5	70.6	72.5	19.3	18.6	20.0
Total	1795.9	1772.1	1819.7	100.0	-	-	221.5	216.2	226.8	100.0	-	-	12.3	12.1	12.6

^1^ NOVA 1: unprocessed and minimally processed foods; NOVA 2: processed culinary ingredients; NOVA 3: processed foods; NOVA 4: ultra-processed foods.

**Table 3 nutrients-14-00708-t003:** Indicators of the dietary content in free sugars across quintiles of ultra-processed food consumption (% total energy intake) among Canadians 2 years or older (*n* = 20,103).

	Relative Energy Intake from Ultra-processed Foods (% Total Energy Intake)			Relative Energy Intake from Free Sugars (% Total Energy Intake from Free Sugars)			Indicators							
							Participants with More than 10% of Total Energy Intake from Free Sugars				Participants with More than 5% of Total Energy Intake from Free Sugars			
		Range			95% CI				95% CI				95% CI	
Quintiles	Mean	Min.	Max.	Mean	Lower Limit (LL)	Upper Limit (UL)	Prevalence	PR ^†^	LL	UL	Prevalence	PR ^†^	LL	UL
Q1	19.4	0.0	29.4	8.1	7.8	8.5	28.4	Ref.	-	-	65.3	Ref.	-	-
Q2	35.8	29.4	42.0	10.9	10.5	11.3	50.1	1.80	1.60	2.02	81.8	1.25	1.18	1.32
Q3	47.4	42.0	52.9	12.7	12.2	13.2	61.5	2.11	1.90	2.35	87.6	1.32	1.23	1.43
Q4	58.9	52.9	65.3	14.5	14.1	15.0	65.5	2.25	2.04	2.48	90.9	1.37	1.31	1.44
Q5	76.4	65.3	100.0	16.8	15.8	17.8	73.1	2.48	2.22	2.76	91.7	1.38	1.31	1.46
Total	45.7	-	-	12.3	12.1	12.6	53.5	-	-	-	82.1	-	-	-

^†^ Prevalence ratio (PR) adjusted for sex, age, household income, household educational attainment, and immigrant status (*n* = 19,962).

## Data Availability

Publicly available datasets were analyzed in this study. This data can be found here: https://www.canada.ca/en/health-canada/services/food-nutrition/food-nutrition-surveillance/health-nutrition-surveys/canadian-community-health-survey-cchs/2015-canadian-community-health-survey-nutrition-food-nutrition-surveillance.html (accessed on 20 December 2021).

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
