# Peer review of "Consumption of Ultra-Processed Foods Is Associated with Free Sugars Intake in the Canadian Population"

_nutrients, 2022, doi:10.3390/nu14030708_

Round 1

Reviewer 1 Report

Using the 2015 CCHS-Nutrition survey data, this study investigates the association between UPF consumption and free sugar intake in the Canadian population. They found that UPF consumption was associated with free 2 sugars intake in the Canadian population.

Methods:

The authors should provide more descriptions on how the association between UPF consumption and free sugars intake was assessed. The Balanced Repeated Replication method is not something widely used, so a lot of readers might not be familiar with the methods. A little more explanations might help the readers better make sense of the results presented in Table 2.

How did the authors handle missing data? More details need to be provided, including % cases with missing data, any imputations done, and any potential biases...

Results

For results in Table 1 and 2, it might be more appropriate to use upper limit (UL) and lower limit (LL) for the 95% confidence interval, instead of max and min. 

Has there been any consideration for subgroup analysis, by age, gender, and race/ethnicity? This should be discussed as potential limitation or future research directions.

Discussions:

The authors noted in the methodology section that the data excluded those who lived on reserves and other Indigenous settlements, in some remote areas. These are very likely to be individuals with low socioeconomic status, medically underserved, with poor access to fresh food. A brief discussion on whether the same association might be observed for these people, and/or what future research should do, would be appropriate.

There is a recent study detailing some issues with the NOVA classification, specifically regarding the definitions of UPT. The authors might consider discussing the use of NOVA classification in this context. If Petrus et al were correct, how does that affect the estimates of this current study? Overestimation or underestimation? Would it affect the associations examined?   Petrus, R. R., do Amaral Sobral, P. J., Tadini, C. C., & Gonçalves, C. B. (2021). The NOVA classification system: A critical perspective in food science. Trends in Food Science & Technology116, 603-608.

Reviewer 2 Report

It is very interesting study with a relevant theme in the context of public health.

I suggest clarifying the study population, as it includes different age groups, but does not detail how each one was analyzed. I believe this is the biggest limitation of the study.

Suggestions:

1) Despite the age control in the analysis, this fact deserves an expanded discussion by age group. As well as including the characterization of the sample.

2)To explain the information: “For children aged 1 to 5 years, interviews were conducted with a proxy only, line 85.

3) It is necessary to include the study flowchart and data on the sociodemographic characteristics of the population.

 4) The methodology must contain whether the recommendation of the 5 steps for 24-hour recall collection was followed.

5) To include a flowchart of the sample, acoording to Strobe checklist.

Round 2

Reviewer 2 Report

The authors answered all requests clearly in the text, and answered the questions properly. Therefore, I am satisfied with the improvement of the manuscript.